# Sub-5 nm AFM Tip Characterizer Based on Multilayer Deposition Technology

Ziruo Wu [1,2,3,4,5], Yingfan Xiong [1,2,3,4,5], Lihua Lei [6], Wen Tan [1,2,3,4,5], Zhaohui Tang [1,2,3,4,5], Xiao Deng [1,2,3,4,5,*], Xinbin Cheng [1,2,3,4,5] and Tongbao Li [1,2,3,4,5]

1   Institute of Precision Optical Engineering, Tongji University, Shanghai 200092, China
2   MOE Key Laboratory of Advanced Micro-Structured Materials, Tongji University, Shanghai 200092, China
3   Shanghai Frontiers Science Center of Digital Optics, Tongji University, Shanghai 200092, China
4   Shanghai Professional Technical Service Platform for Full-Spectrum and High-Performance Optical Thin Film Devices and Applications, Tongji University, Shanghai 200092, China
5   School of Physics Science and Engineering, Tongji University, Shanghai 200092, China
6   Shanghai Institute of Measurement and Testing Technology, Shanghai 201203, China
\*   Correspondence: 18135@tongji.edu.cn

**Abstract:** Atomic force microscope (AFM) is commonly used for three-dimensional characterization of the surface morphology of structures at nanoscale, but the "Inflation effect" of the tip is an important factor affecting the accuracy. A tip characterizer has the advantages of in situ measurement, higher accuracy of probe inversion results, and relatively simple fabrication process. In this paper, we developed a rectangular tip characterizer based on multilayer film deposition technology with protruding critical dimension parts and grooves parts. And the tip characterization is highly consistent across the line widths and grooves, and still performs well even in the sub-5 nm line width tip characterizer. This indicates that tip characterizers produced by this method can synergistically meet the combined requirements of standard rectangular structure, very small line edge roughness, very small geometry dimension, and traceable measurements.

**Keywords:** atomic force microscope; tip characterizer; multilayer deposition; critical dimension

## 1. Introduction

Nanometer length metrology is instrumental in the development of advanced nanomaterials and devices. Atomic force microscope (AFM) is commonly used for three-dimensional characterization of the surface morphology of structures at nanoscale [1], and the subsequent development of CD-AFM [2] is also useful for characterizing samples with small linewidths. Compared with the averaging measurement of scattering [3,4], AFM measurements can provide a very accurate information about the local structure. In recent years, AFM has also proven to be an important support tool for photonics research [5,6]. For example, the AFM-IR technique [7] that combines the high spatial resolution of AFM with the chemical identification capability of infrared (IR) spectroscopy has the ability of achieving submicrometric physical-chemical analyses. And the AFM-IR is also a non-linear interaction between the AFM tip and the sample surface. Therefore, the measurement accuracy of AFM is important for the subsequent development of nanomanufacturing and advanced analyses for materials. However, the distortion of AFM imaging is serious when the tip size is close to the sample surface topography [8], and the "Inflation effect" of the tip is an important factor affecting the accuracy of AFM measurement, which is a problem that must be faced and solved when AFM achieves "True 3D" measurement.

Villarrubia. J. S's mathematical morphology-based erosion algorithm proposed that the ability to accurately characterize the AFM scanning tip is the key to reconstruct and optimize the AFM imaging results [9]. There are various methods commonly used for tip characterization, such as direct imaging using SEM [10,11], blind reconstruction [9–16], and

tip characterizers [17]. Direct imaging of the probe by SEM suffers from the inability to measure in situ, time-consuming, unguaranteed integrity, and possible contamination [18]. The blind reconstruction method requires a more demanding sample, which needs to have a sharper bump, and the blind reconstruction method has the disadvantages of being affected by noise [12,14]. Compared with the previous two inversion means, the tip characterizer has the advantages of in situ measurement, higher accuracy of probe inversion results, and relatively simple fabrication process.

The characteristics of the tip characterizer should meet the following points. (i) A simple and standard profile shape that facilitates probe inversion based on the scanned profile. Among them, the good vertical structure of rectangular shape and the left and right top angles are 90° can effectively simplify the reconstruction process and improve the inversion accuracy. (ii) The tip characterizer structure should have a low line edge roughness and good inter-sample consistency. Since the subsequent TEM calibration process requires destructive slicing of the sample, which prevents further transfer of the tip characterizer properties [19], more than one comparable tip characterizer sample needs to be scanned in the inversion process. In addition, if tip characterizer batch production can be performed and good consistency can be maintained, it will be of great value to improve the measurement consistency between different AFM instruments. (iii) The characteristic structural part of the tip characterizer should be small enough. The reduction of the characterizer's own scale is of great value for the AFM tip inversion. In the calibration process of AFM tip, the scanning image is the result of the interaction between the tip characterizer and the scanning tip [10]. The smaller the size of the line width part of the calibrator, the more accurate the tip shape obtained by inversion. (iv) The tip characterizer's own structure should be measured accurately. Measurement traceability is the basis of measurement accuracy. In 2019, the lattice spacing of silicon has been utilized as secondary realizations of the length unit at the nanometer and sub-nanometer scale [20]. This provides the basis for establishing traceable measurements for very small tip characterizers. Therefore, the key to developing a tip characterizer is to fully consider the above-mentioned multiple aspects of tip characterizer feature requirements and to achieve synergistic optimization of tip characterizer functions [19].

Motivated by the above technical requirements for tip characterizers in the field of AFM measurements, we developed a rectangular tip characterizer based on multilayer film deposition technology. And the tip characterizers produced by this method can synergistically meet the combined requirements of standard rectangular structure, very small line edge roughness, very small geometry dimension, and traceable measurements. We verified the function of surface profile information and depth-to-width ratio characterization of the multilayer film tip characterizer for AFM scanning tips. The experimental results show that the multiscale tip characterizer has good characterization consistency and the sub-5 nm rectangular tip characterizer also still maintains good tip characterization results. The above study confirms that the multilayer film tip characterizer is an effective way for AFM characterization.

## 2. Materials and Methods

### 2.1. Fabrication of Tip Characterizer

In this paper, the tip characterizer is fabricated using a multilayer film deposition technique, which takes advantage of the homogeneity of its film layer growth, and the thickness of the deposited film layer is directly translated into the structural width of the tip characterizer during fabrication [21]. Specifically, the multilayer film structure [22] with different thicknesses is designed and prepared according to the difference of deposition rates by using ion beam sputtering [23–25]. And the multilayer deposition technology is very useful for *X* ray focusing and high-power laser systems [26,27]. This layer-stacked film structure can maintain its stability and good uniformity between the film layers in a wide range, and two wafers with the same deposited film layers are subsequently bonded, cut, ground and polished, and the desired tip characterizer is prepared by selecting the wet

etching method. The fabrication process is shown in Figure 1. In the above process, the maintenance of the rectangular characteristics of the tip characterizer profile is achieved by vertical cutting and grinding.

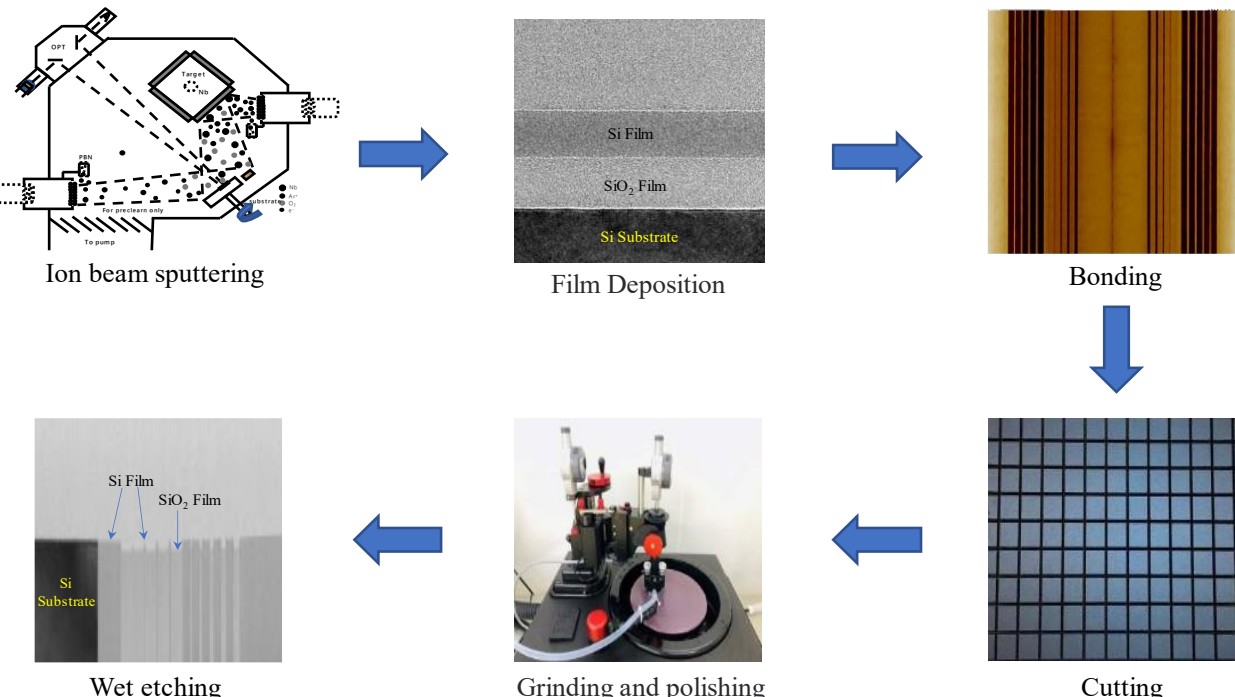

**Figure 1.** Fabrication process of tip characterizer based on multilayer deposition technique.

It should be noted that here we use silicon wafer bonding [28] instead of gluing to bond two samples with the same deposited film layer structure together, which solves the problems of contamination of the glued jointing layer. In addition, the presence of the bonding layer also facilitates the researchers to perform traceable measurements of the tip characterizer structure. After the samples were cut and etched after silicon wafer bonding, the measurements of the linewidth structure were more accurate and consistent among the samples, and the linewidth part of the cross section was rectangular.

We designed and manufactured the multilayer film tip characterizer by selecting silicon and silicon oxide as the material pair for subsequent traceability measurement characterization. Figure 2a shows the design of the tip characterizer, where the linewidths of the protruding parts are 5 nm, 10 nm, 15 nm, and 20 nm, and the widths of the recessed grooves are 10 nm, 20 nm, 30 nm, 40 nm, and 50 nm, respectively. Figure 2b shows the structure of the tip characterizer prepared according to the above preparation procedure with symmetric distribution. The subsequent TEM measurements show that the minimum nominal linewidth value is slightly less than 5 nm, so this preparation process can stably achieve the fabrication of sub-5 nm tip characterizer structure.

## 2.2. Traceable Measurements of Tip Characterizer Structure

The process of calibrating a tip characterizer for probe morphology involves the accurate measurement of the calibrator's own structure, making it particularly necessary to calibrate the linewidth portion of the tip characterizer. In the 2019 revision of the SI, the major addition is to put forward the secondary methods of realizing the meter for dimensional nanometrology based on Silicon lattice parameter, which has been already utilized to measure the step height, critical dimension and nano-gratings [29,30]. Based on precise *X*-ray measurements, the spacing of undoped Si(111) is 313.56011(17) pm. Since the cell structure of single-crystal silicon is cubic diamond type, the use of Si(111) as a measurement standard is essentially equivalent in traceability to Si(220) [31], which is

commonly used internationally. In the measurement calibration of the tip characterizer linewidth samples, we calibrated them with a TEM calibrated by a silicon lattice, and achieved the purpose of traceability calibration.

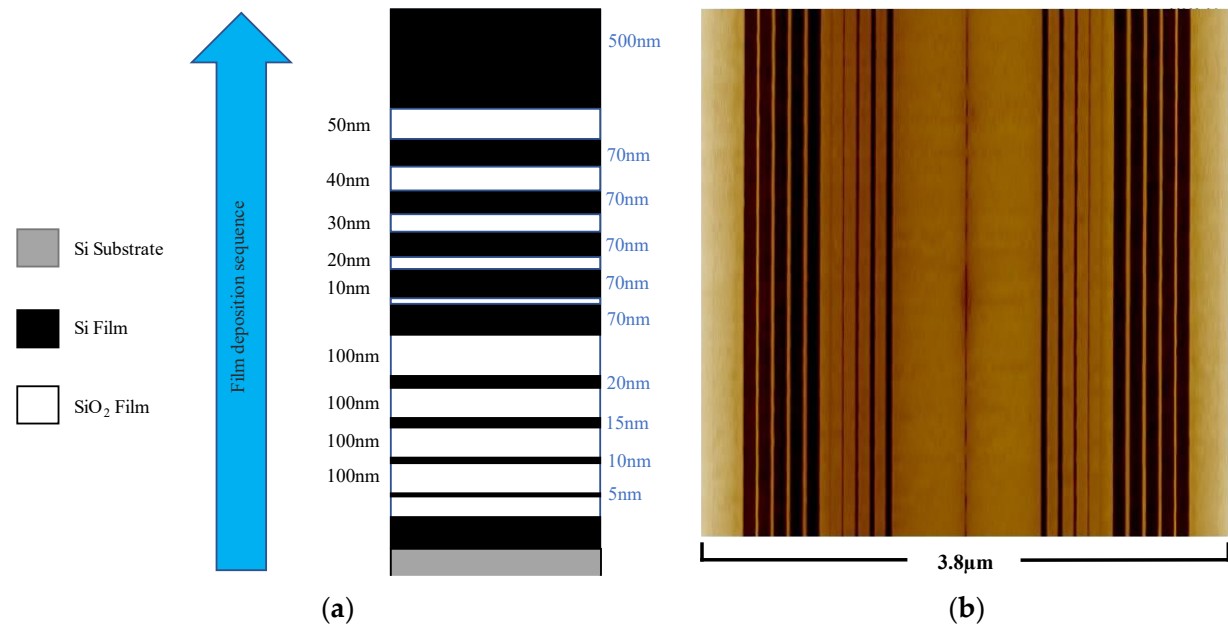

(a)　　　　　　　　　　　　　　　　　　　　(b)

**Figure 2.** Multi-structured tip characterizer:(**a**) film layer design; (**b**) AFM image.

### 2.3. Principle of AFM Tip Structure Inversion Method Based on Tip Characterizer

The principle of tip characterizer to reconstruct the AFM tip structure is schematically shown in Figure 3 below. The AFM tip size cannot be ignored, and furthermore, when the tip scans the sample from the left side of the tip characterizer, the information on the right-side surface of the tip is left on the scanning contour. And when the tip scans the top-of-the-line width back to the bottom from the right line edge, the information on the left side surface of the tip is left on the contour, as shown schematically below inf Figure 3a.

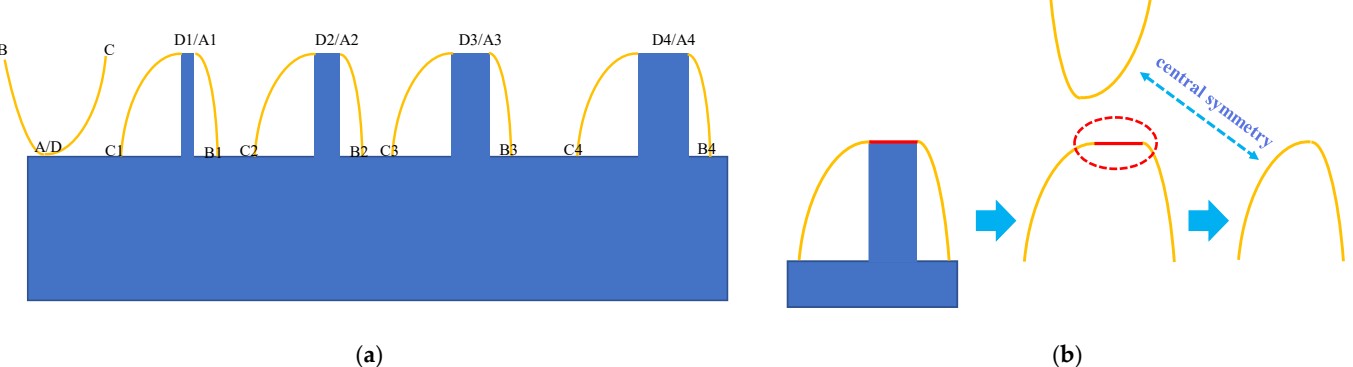

(a)　　　　　　　　　　　　　　　　　　　　(b)

**Figure 3.** The principle of tip characterizer to reconstruct the AFM tip structure: (**a**) Imaging process; (**b**) Reconstruction process.

As the tip size cannot be ignored, the result will be a slightly inflated pattern (solid yellow line). Geometrically subtracting this outline from the corresponding line width portion (shown as a solid red line in the Figure 3b) gives a centrosymmetric pattern of the scanning tip about the tip, then the calibration about the tip outline is achieved. This basic profile should be similar for different nominal line widths. The profile calibration diagram is shown in Figure 3b below. It is important to note that the smaller the line width size, the closer the profile of the tip after scanning is to the true surface of the tip, and the better it is

for us to reconstruct the shape of the tip. Compared to a 20 nm linewidth, we believe that a 5 nm linewidth scan profile is more advantageous and credible for inversion of the tip shape, with less inversion error.

The notch portion of the characterizer with different widths is then used to extract the tip depth to width ratio information. We have designed the notches with a groove width of 10 nm, 20 nm, 30 nm, 40 nm and 50 nm, respectively. When the probe is scanned into the notch of width W1, the maximum probe depth is H1, when it is scanned into the notch of width W2, the maximum tip depth is H2, and so on. In this way, a series of tip depth to width ratio information can be obtained, and the schematic diagram is shown in Figure 4 below.

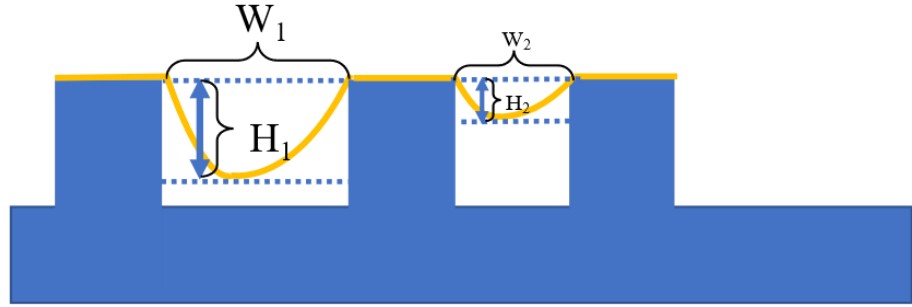

**Figure 4.** The principle of tip characterizer to reconstruct the AFM tip depth to width ratio.

### 3. Results

*3.1. Tip Characterizer Fabrication and Traceable Measurements*

According to the multi-structured tip characterizer film structure design in the previous section, the TEM scan pattern of the overall structure of our completed $Si/SiO_2$ multilayer film grating sample is shown in Figure 5 below. The tip characterizer was fabricated by cutting the whole deposited sample into the same size, so the film structure between the samples is almost the same.

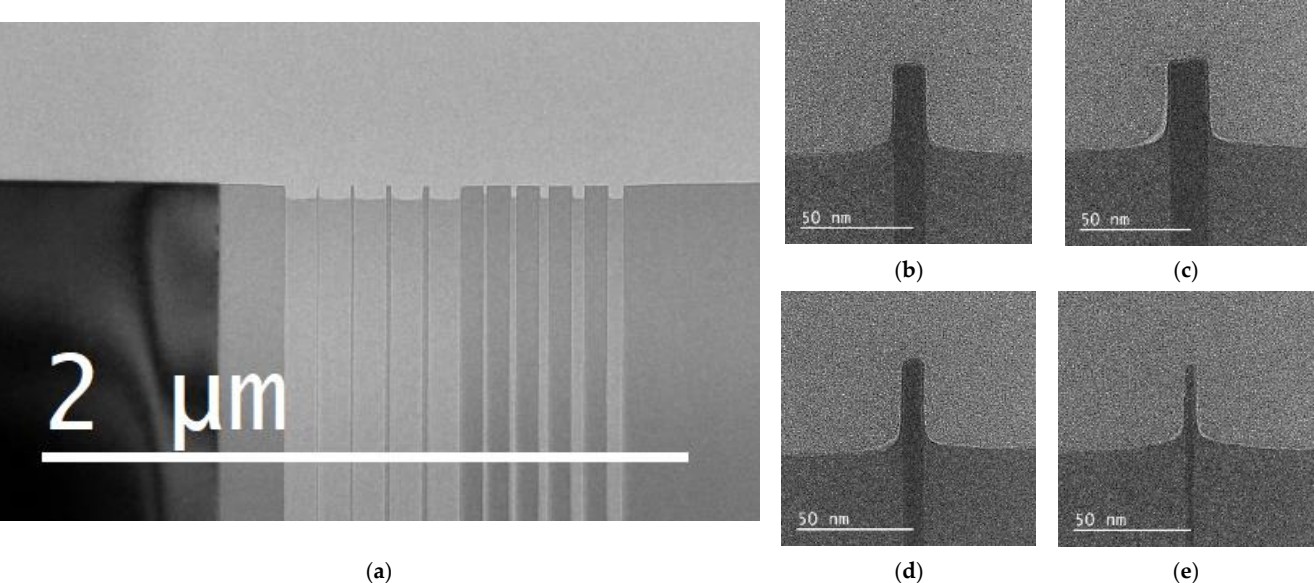

**Figure 5.** TEM image of multi-structured tip characterizer: (**a**) The whole profile of protruding parts and grooves; (**b**) 20 nm protruding CD structure; (**c**) 15 nm protruding CD structure; (**d**) 10 nm protruding CD structure; (**e**) 5 nm protruding CD structure.

It is important to note that the sharpness of the line width section of the tip characterizer will affect the inversion results of the tip, so we have measured the inclination angle

and calculated the roughness of the line width section for each size. The inclination angles are 87.25° on the left side and 87.69° on the right side for the 5 nm line width, 88.67° on the left side and 88.14° on the right side for the 10 nm section, 89.61° and 88.92° on the left side and right side respectively for the 15 nm section, and 89.47° and 89.27° on the left side and right side respectively for the 20 nm section.

According to the traceable measurement method based on the silicon lattice, we first calibrate the TEM using the lattice information of the silicon substrate, followed by imaging measurements of the protruding linewidth structure of the tip characterizer. The measurement results of both sides of the tip characterizer are shown in Figure 6. Here let us assume that one side of the calibrator is the left side, and when it is rotated by 180° after the measurement, that side is the right side. The results of this traceable measurements show that the measured average value of the central linewidth of the protruding linewidth structures with nominal values of 20 nm, 15 nm, 10 nm and 5 nm are 18.91 nm, 14.65 nm, 9.47 nm and 4.68 nm respectively, with a standard deviation of less than 0.5 nm. The above results indicate that the multi-structured tip characterizer have good structural uniformity.

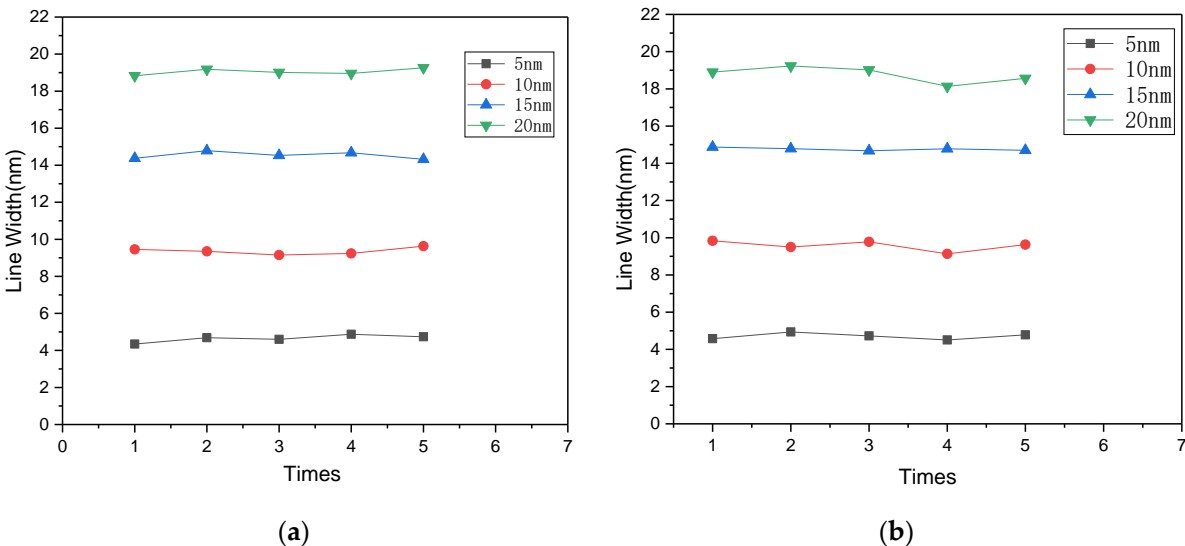

**Figure 6.** Traceable measurement results of middle CD value of multi-structured tip characterizer: (**a**) Left side; (**b**) Right side.

*3.2. AFM Tip Profile Reconstruction Based on Tip Characterizer*

Next, we used the developed tip characterizer to perform tip structure inversion on a commercial AFM probe. The probe model we used is Bruker's RTESPA-300, which is a quadrilateral conical probe with different anterior and posterior angles, and the expected tip radius is 8~12 nm. We define the tip parallel to the cantilever beam direction as the X-direction and perpendicular to the cantilever beam direction as the Y-direction. When scanning, the multi-structured multilayer tip characterizer is first scanned along the X direction, and then the multilayer tip characterizer is rotated 90° to continue scanning along the Y direction. The specific scanning process is illustrated in Figure 7a. The results of a typical scan along the X-direction are shown in Figure 7b.

We randomly selected three profile lines in the Figure 7b for tip structure inversion, and the measured profiles of the 5 nm, 10 nm, 15 nm and 20 nm linewidth tip characterizer are shown in the Figure 8a, and by subtracting the linewidth values from the TEM measurements based on the inversion method above, the AFM probe tip structure are as shown in Figure 8b. From the above characterization process of the AFM probe tip, the following conclusions can be drawn: (1) The structure of the AFM probe tip characterized by different profiles of the same linewidth tip characterizer for each quantity is basically the same, indicating that the linewidth tip characterizer is extremely uniform and has good characterization consistency throughout the region of the quantity. This is consistent

with the morphological results observed in the TEM and AFM images obtained from our measurements. (2) The consistent characterization results of the AFM probe tips by the linewidth tip characterizer of different magnitudes indicate that the linewidth tip characterizers are very accurate in the pre-TEM measurement process, which ensures the consistency of the measurements by the linewidth tip characterizers of different magnitudes; (3) The tip structure results in Figure 8b show slight differences between the anterior and posterior angles of the AFM tip, which is consistent with the probe design provided by the manufacturer, and indirectly proves that the characterization results of the tip characterizer in this paper are valid. Similarly, we performed the same probe characterization and inversion along the Y direction, and the results obtained are shown in Figure 9. This result also demonstrates the consistency and accuracy of the inversion of the tip characterizers.

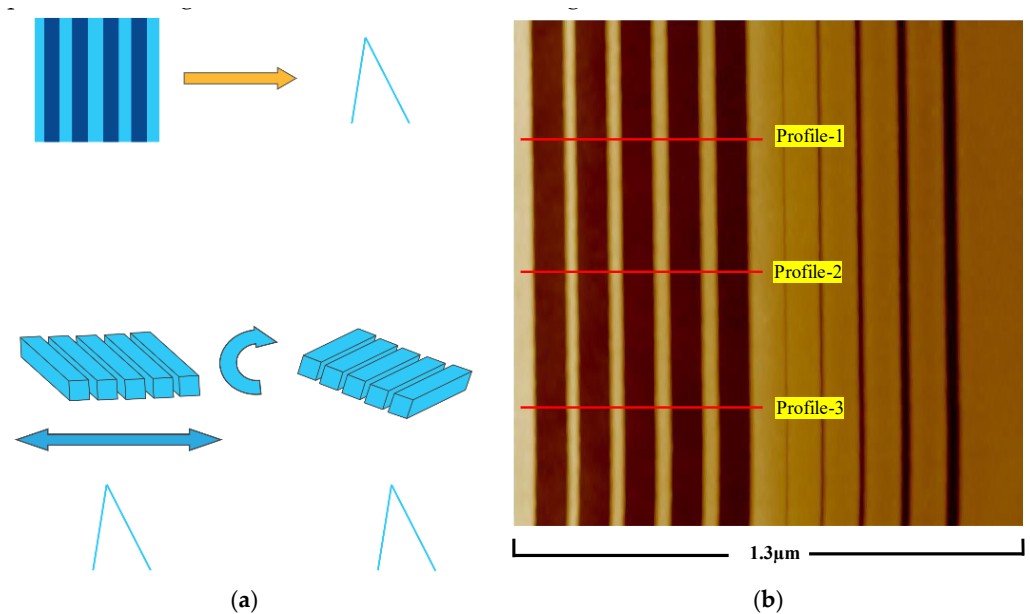

| (a) | (b) |

**Figure 7.** (**a**) Scanning process of tip characterizer with a commercial probe; (**b**) Typical AFM scan results of the tip characterizer.

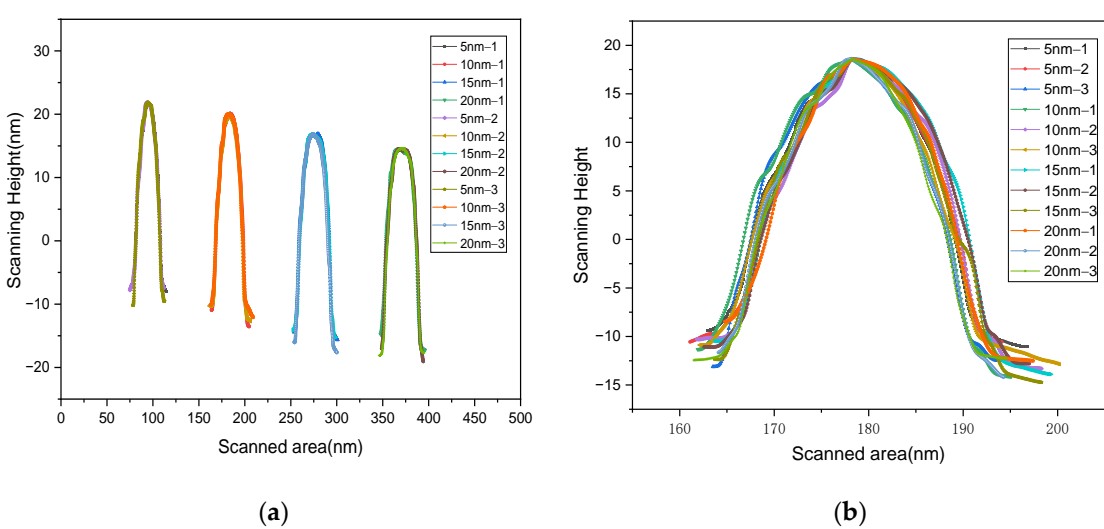

| (a) | (b) |

**Figure 8.** (**a**) Randomly selected measured profiles of the 5 nm, 10 nm, 15 nm and 20 nm linewidth tip characterizer along X direction; (**b**) AFM probe tip structure obtained along X direction by the tip characterizer.

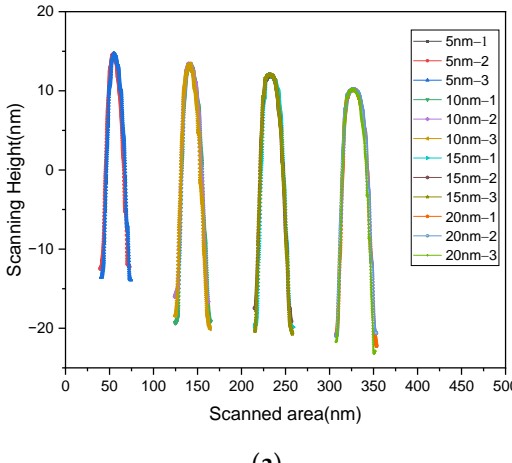

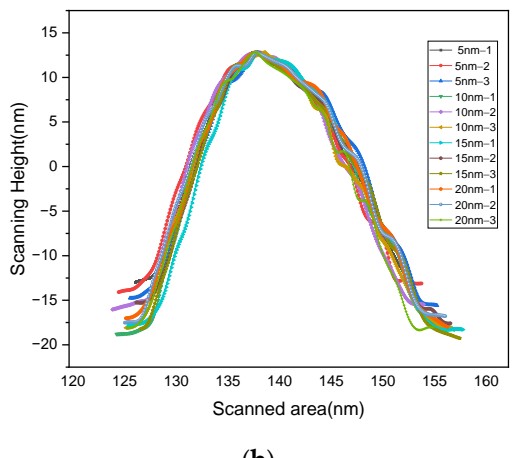

(**a**)

(**b**)

**Figure 9.** (**a**) Randomly selected measured profiles of the 5 nm, 10 nm, 15 nm and 20 nm linewidth tip characterizer along Y direction; (**b**) AFM probe tip structure obtained along Y direction by the tip characterizer.

### 3.3. AFM Tip Depth to Width Ratio Characterization Based on Tip Characterizer

The linewidth of the tip characterizer is used to extract the profile of the tip, while the grooves' part can be used to extract the information of the tip's aspect ratio. When the width of the groove is small, the location that the probe can go down to is also very limited, and as the groove widens, the area that the tip can scan to is gradually deepened. A series of depth-to-width ratio analyses were performed. Figure 10 demonstrates the scanning AFM image of tip characterizer along X direction and Y direction. This time we focus on the right part of each AFM image, which is the groove part of the tip characterizer. As can be seen in the Figure 10, the structure of the grooves is flat and uniform. This indicates that the silicon oxide film layer and the silicon film layer are equally uniform and consistent. We randomly selected groove profiles line in Figure 10 for comparative analysis, as shown in Figure 11a. By placing the height and depth information in the same figure it is found that as the trench width increases, basically the depth that the tip can reach also increases. However, the depth obtained did not increase significantly when the trench width was increased from 40 nm to 50 nm. We speculate that this is most likely due to the tip touching the bottom of the trench. Based on the principle of the inverse depth-to-width ratio of the tip characterizer described above, we calculated the information corresponding to the depth and width of the tip that can be inverted from the above image, as shown in Figure 11b.

To address the above-mentioned characterization inaccuracy due to probe bottoming, we increased the depth of the grooves by increasing the wet etching time. We increased the etching time from 1.5 min to 2 min, 3 min, and 4 min respectively during the processing of the other three tip characterizer. The information on the depth-to-width ratio of the probes obtained from the three new tip characterizers is shown in Figure 12. As can be seen from the curves in Figure 12, the bottoming out problem has been avoided at 3 and 4 min of etching, and therefore the information of the inverted aspect ratio agrees along both X and Y direction. At the same time, it is worth noting that, overall, there is a difference in the aspect ratio of the tip in the X and Y directions, which is also reflected by the tip characterizers. These results show that the multilayer-based trench-type tip characterizers can provide a better characterization of the probe tip.

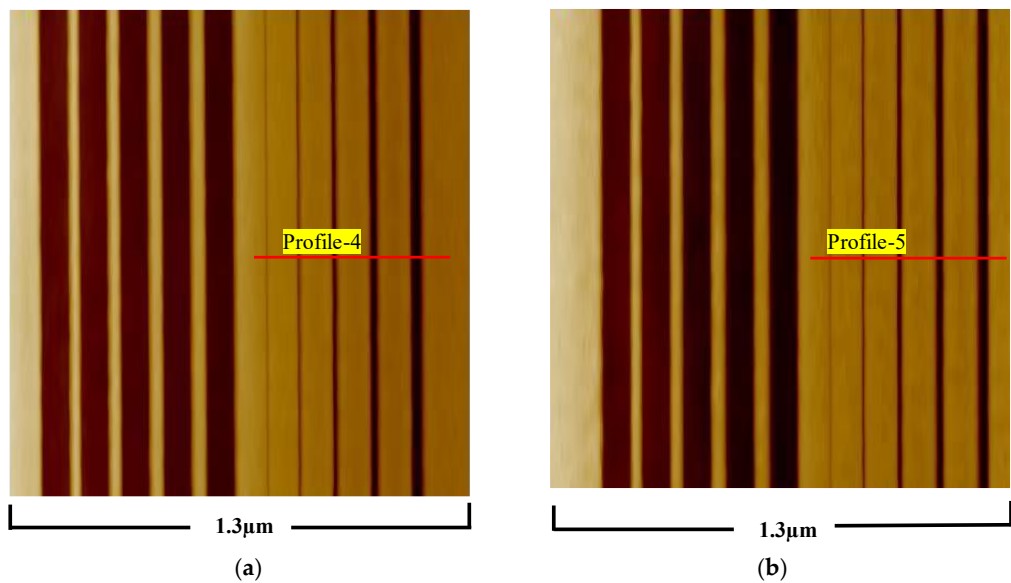

**Figure 10.** Scanning AFM image of tip characterizer along (**a**) X direction and (**b**) Y direction.

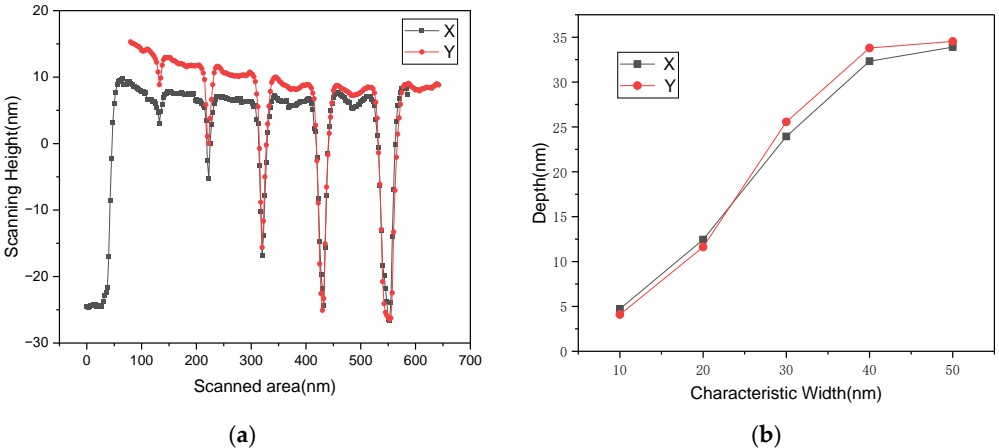

**Figure 11.** (**a**) Scanning AFM image profile of grooves part in tip characterizer; (**b**) Information corresponding to the depth and width of the tip.

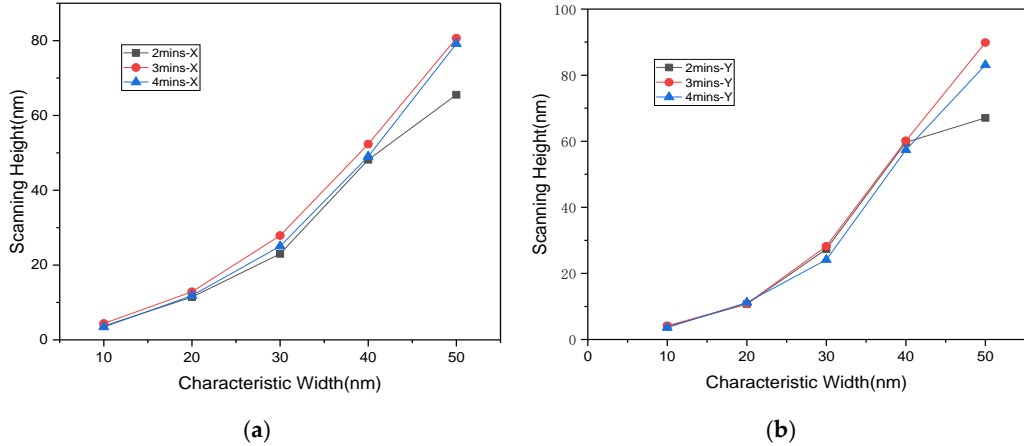

**Figure 12.** Information corresponding to the depth and width of the tip based on tip characterizers of different etching time: (**a**) along X direction; (**b**) along Y direction.

## 4. Discussion

From the above, the multilayer film-based tip characterizer can achieve inversion of the probe tip profile and depth-to-width ratio with good agreement. However, it is also important to note that there are several aspects of this method that need to be considered.

The first is about the methodological aspects of the development of the tip characterizer. A precise tip characterizer requires not only a small geometric scale, but also an actual structure that is consistent with the ideal structure. Through the research in this paper, it is found that for tip characterizer above 10 nm, a better rectangular structure can be guaranteed because the multilayer film technology is converted to line width after the film. However, when it reaches 5 nm, the pressure of the bonding process tends to cause distortion of the linewidth structure, resulting in poor structural consistency. Therefore, the subsequent processing of linewidth calibrators by methods such as direct electron beam writing may be an alternative option.

By the way, we have measured the line width of the characterizer in this paper and the results show that the inclination angle of the line width of each size is closer to 90°, which provides a strong basis for accurate results in subsequent experiments, but we need to note that due to process problems, there is always a deviation from the perfect 90° in the fabrication within the micro-nano scale, and how to approximate 90° will be the focus of our subsequent research.

In the second aspect, we found that the raised and notched tip characterizers have different advantages and disadvantages in inverting the tip structure, and they can meet different measurement needs respectively. The raised tip characterizer is mainly used for tip profile inversion, where the scanned profile of the AFM tip is processed to obtain the tilt profile information, angle information, and depth-to-width ratio information of the tip. However, the width of the tip characterizer needs to be subtracted when performing the inversion, which is more complicated to calculate, more parameters, and less efficient. The notch part can give the aspect ratio information directly and quickly, and the inversion is efficient for the needs of aspect ratio only. Therefore, the combination of the two can be satisfied for different tip requirements, which can be used as a structural reference for the subsequent design of tip characterizers for researchers

In the third aspect, when using TEM to measure the multilayer film tip characterizer, the presence of oxide layers on both sides of the raised line width structure makes it extremely important to accurately determine the edges of the film layers. This is also an important factor affecting the accuracy of the method. The development and research of related line edge detection algorithms may be effective in terms of improving the accuracy of the film edge. The second aspect is that the raised line-width tip characterizer structure at sub-5 nm is not only easier to collapse, but also differs from the expected rectangular structure. It will be particularly important to optimize the preparation process and improve the yield of the sub-5 nm tip characterizer in the future.

## 5. Conclusions

This paper presents a $Si/SiO_2$ multilayer film tip characterizer based on ion beam sputtering and film deposition. The tip characterizer has the advantages of simple rectangular cross-sectional profile, low line edge roughness, good inter-sample agreement, high stability of the film layer, and small characteristic line widths that can be easily and accurately measured by Si lattice spacing. The experimental results show that the tip characterizer satisfies the various needs of the tip in the surface profile calibration, and the inverse structure of the tip is highly consistent across the line widths and still performs well in the sub-5 nm line width tip characterizer, which fully demonstrates the feasibility of the multilayer method. In terms of tip aspect ratio extraction, the notch part of the tip characterizer can also extract the aspect ratio information of the feature location more effectively, and the notch part where the probe height cannot be extracted effectively can be solved by increasing the etching time, and the aspect ratio in the Y-direction is larger than that in the X-direction when the probe width is 40 nm and 50 nm. The fabrication of sub-5

nm linewidth multilayer film tip characterizer makes the calibration of AFM scanning tip more accurate, which is beneficial to the subsequent optimization of AFM scanning images to "True 3D" measurements.

**Author Contributions:** Conceptualization, Z.W.; Data curation, Y.X. and W.T.; Formal analysis, Y.X.; Investigation, L.L., W.T. and Z.T.; Methodology, Z.W.; Resources, L.L., X.C. and T.L.; Supervision, X.D., X.C. and T.L.; Writing—original draft, Z.W.; Writing—review & editing, X.D. All authors have read and agreed to the published version of the manuscript.

**Funding:** This research was funded by Special Development Funds for Major Projects of Shanghai Zhangjiang National Independent Innovation Demonstration Zone (ZJ2021-ZD-008), National Natural Science Foundation of China (Grant No. 62075165), Program of Shanghai Academic Research Leader (21XD1425000), Shanghai Municipal Science and Technology Major Project (2021SHZDZX0100) and the Fundamental Research Funds for the Central Universities, Opening Fund of Shanghai Key Laboratory of Online Detection and Control Technology (ZX2020101).

**Institutional Review Board Statement:** Not applicable.

**Informed Consent Statement:** Not applicable.

**Data Availability Statement:** Not applicable.

**Conflicts of Interest:** The authors declare no conflict of interest.

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
