# Peer review of "Sub-5 nm AFM Tip Characterizer Based on Multilayer Deposition Technology"

_photonics, doi:10.3390/photonics9090665_

Round 1
Reviewer 1 Report
This work may be published in appropriate journals. Suitability to photonics is questionable.
I do not recommend publication of this work in Photonics.
1) Although the prepared structures may be useful photonics in future, results presented in
this results are far from photonic research. This work may be published in the other journals.
2) fabrication methods are represented with cartoons, which are not well understandable.
Author Response
Dear Reviewers.
Thank you very much for reviewing our manuscript and providing guidance during your busy schedule, which will be of great importance to our future experimental research. We hope that the revised article will meet your requirements and that our independent opinion will be accepted by the reviewers.
The specific responses to you are as follows.
Reply to reviewer 1:
Comment 1: Although the prepared structures may be useful photonics in future, results presented in these results are far from photonic research. This work may be published in the other journals.
Reply:
The reviewers' comments are very pertinent.
Firstly, this paper describes the use of a multilayer film linewidth tip calibrator to accurately characterize the surface profile of a tip, which is of great importance not only for AFM precision measurements of micro and nano structures, but also for cutting edge photonics or spectroscopy research methods such as AFM-IR, a technique that uses the tip of an AFM scanning tip to locally detect thermal expansion from infrared radiation absorption in a sample. AFM-IR is a technique that uses the tip of an AFM scanning tip to locally detect the thermal expansion generated by the absorption of infrared radiation in a sample, using the high resolution of AFM and the chemical recognition capability of infrared spectroscopy to achieve sub-microscopic analysis. Therefore, we believe that the inversion of the scanning tip shape is a common need and that this method is supportive for future research in photonics.
Secondly, the target journal for this paper is the Optical Thin Films section of the Special Issue of photonics, where the research method is based on multilayer films, which is perfectly in line with the technical theme of the issue.
In summary, our submission to this journal is not only a good fit with the issue theme of the journal, but also the supporting technology that underpins photonics research, and therefore we believe it is appropriate. To address this issue raised by the reviewer, we have added a description of the need for AFM tip calibration in AFM-IR, which makes the publication of this paper more reasonable.
Comment 2: fabrication methods are represented with cartoons, which are not well understandable
Reply:
We are very grateful to the reviewers for pointing out the deficiencies in the presentation of the images. In order to facilitate the understanding of the reviewers and readers and to make the images more professional, we have optimized them and hope that they will meet the reviewers' requirements.
The figure is demonstrated in the attached file.
Figure 1. Fabrication process of tip characterizer based on multilayer deposition technique

Reviewer 2 Report
This is a great manuscript exploring and discussing a novel multilayer film tip characterizer targeting for Sub-5 nm AFM tip. The performance of the characterizer was validated by the commercial AFM tip surface profile calibration. This described method has great potential to be applied in AFM optimization and real 3D scanning. The manuscript is well organized with the clear introduction, well designed experiments, detailed discussion and necessary references. As the reason listed above, I suggest publishing this manuscript with some spelling check.
Author Response
Reply to reviewer 2:
Thank you very much for reviewing and providing guidance on our manuscript during your busy schedule. Your high evaluation of the article has also given us confidence in its successful publication. In response to your comments, we have thoroughly checked the article and corrected and marked some inaccuracies: for example, changing "top of the line" to "top-of-the-line", besides, the word "probe calibrator" has been changed to "tip characterizer", and there are a number of other additions and deletions, which we will not list here for your convenience.
Secondly, we are honored that the article has been so well received and we thank you for your kind words. Our probe inversion studies based on multilayer film linewidths will be valuable and supportive for the future accurate characterization of AFM and the future development of AFM-IR. We therefore hope that this article will be published in photonics with your support, and that it will be brought to the attention of more research colleagues, thus contributing to this research direction.

Reviewer 3 Report
It is quite meaningful to characterize structure with AFM needle tip. As described in the manuscript, it is more accurate than the blind reconstruction and SEM imaging. But I have several questions.
1. I think the accuracy of the calibration to the needle tip is more related to the sharp level of the characterizer. I think it is necessary to emphasize the sharpness and dip angle of the characterizer.
2. how to get the tip morphology, minus the line width? What line width of 5nm, 10nm, 15nm can give you compared to that of 20nm? What is the benefits using smaller line width?
3. Using rectangular characterizer can obtain the depth to width ratio of the needle tip, then why some groove structure is needed? What are the advantages groove can give?
Something more needs to pay attention:
1. In page 157, “by design, we have designed the notches with a gradient of 10nm to 50nm, and the gradient is 10nm”. There maybe something wrong here.
2. In figure 6, what does left side and right side means?
Author Response
Reply to reviewer 3
Dear Reviewers.
Thank you very much for reviewing our manuscript and providing guidance during your busy schedule, which will be of great importance to our future experimental research. We hope that the revised article will meet your requirements and that our independent opinion will be accepted by the reviewers.
The specific responses you are as follows.
Reply to reviewer 3:
Comment 1: The accuracy of the calibration of the needle tip is more related to the sharpness of the characterizer. The sharpness and inclination of the characterizer should be emphasized in the text.
Reply:
First, we strongly agree with the reviewers' comments. Assuming that we have a tip characterizer with a perfectly rectangular and sharp cross section of arbitrary line width, the closer the tip shape obtained by the inversion process will be to the actual value.
However, we have a different understanding on how to accurately calibrate the tip shape. We believe that whether the tip characterizer can accurately calibrate the tip of the scanning tip is reflected in two aspects; on the one hand, the size of the line width section of the characterizer, the smaller the size of the rectangular line width section, the closer the scanning profile is to the shape of the tip itself, and the error between the inversion result and the actual tip epoch shape is smaller, and the more precise the calibration is. The smaller the size of the rectangular line width, the closer the scan profile is to the shape of the tip itself, the smaller the error between the inversion result and the actual tip epoch shape, the more precise the calibration. On the other hand, whether the linewidth structure is standard (small or no difference between the actual size and the TEM characterization result) and whether the structure can be accurately characterized is also an important issue affecting the calibration accuracy.
This issue is addressed in the discussion section of the paper, and in the text we show how difficult it is to meet both the small size and the standard shape of the characterizer by demonstrating the variation of the 20 nm to 5 nm linewidth structure. In the future, we will consider direct electron beam writing in the hope of eventually achieving a more standardized state.
Comment 2: how to get the tip morphology, minus the line width? What line width of 5nm, 10nm, 15nm can give you compared to that of 20nm? What is the benefits using smaller line width?
Reply:
First of all, we are very grateful to the reviewers for asking questions about this process, and we also feel that a thorough description of the inversion principle is necessary, and the additional description has been added in the text.
First of all, as the tip size cannot be ignored, the result will be a slightly inflated pattern (solid yellow line). Geometrically subtracting this outline from the corresponding line width portion (shown as a solid red line in the figure 1) gives a centrosymmetric pattern of the scanning tip about the tip, then the calibration about the tip outline is achieved. This basic profile should be similar for different nominal line widths. The profile calibration diagram is shown in Figure 1 below.
The figure is demonstrated in the attached file.
Figure 1. The principle of tip characterizer to reconstruct the AFM tip structure
To address the second issue, we believe that the inversion of the tip morphology using the 20 nm, 15 nm, 10 nm, and 5 nm line width fractions, respectively, shows from the article that the inverse profile of the tip basically matches, indicating a high degree of consistency of the experimental measurements. Secondly, it should be noted that the tip characterizer we developed has other functions in addition to responding to the tip surface topography and aspect ratio information. For example, if the width of the tip can be obtained by scanning the sidewall (Dai, G.L., et al, Opt. Eng. 2016, 55, 7.) (Kwak, G.Y., et al, Applied Surface Science 2021, 565.), related research work is underway and we have performed the experimental data in the paper A simple processing was performed, and a plot was drawn with the measurement map of the TEM as the horizontal coordinate and the half-height width of the line width measurement contour as the vertical coordinate, as shown below, and the width of the tip (Y-axis intercept) can be determined by calculating the slope and intercept.
In response to the reviewer's last question, what is the benefit of using a smaller line width.
We believe that the AFM scan results are the result of the interaction between the two, the scanning tip and the sample. During normal scanning, if the scanning tip size of the AFM is small enough, the smaller the expansion effect will be and the scan results will be accurate. When the linewidth sample size of the characterizer is small enough to reach much smaller than the tip size, we generally regard the rectangular linewidth part of the characterizer as the tip, and the whole process is equivalent to scanning the AFM scanning tip with a smaller tip, so the smaller the linewidth size is, the more accurate the surface morphology of the tip obtained.
Comment 3: Using rectangular characterizer can obtain the depth to width ratio of the needle tip, then why some groove structure is needed? What are the advantages groove can give?
Reply:
Many thanks to the reviewer for asking such a professional question about the design of the characterizer, to which our answer is as follows. The raised line width parts and groove parts of the characterizer are designed to meet different measurement needs. The raised line width parts is mainly used for the profile inversion of the tip. This part can obtain the tilt profile information, angle information, and aspect ratio information of the tip by processing the scanned profile of the AFM tip. However, the width of the tip characterizer needs to be subtracted when performing the inversion, which is more complicated to calculate, more parameters, and less efficient. While the groove parts can give the aspect ratio information directly and quickly, which is efficient for the needs of only aspect ratio. Therefor the combination of the two can be satisfied for different tip requirements. We illustrate this in the discussion section of the paper.
Comment 4: In page 157, “by design, we have designed the notches with a gradient of 10nm to 50nm, and the gradient is 10nm”. There maybe something wrong here.
Reply: This sentence does have some logical and expressive problems, and we have changed the sentence to read: We have designed the notches with a groove width of 10nm, 20nm, 30nm, 40nm and 50 nm, respectively
Comment 5: In figure 6, what does left side and right side means?
Reply: I apologize for not visualizing the left-right side in the article. We randomly select the line width section and the notch section on one side of the tip characterizer to measure and named it the left side, then rotate it 180° in the horizontal plane and take the same measurement, then that measurement will be recorded as the right side. The details have been added to the paper.

Reviewer 4 Report
The work of Wu et al. reports on a dedicated structure for the calibration of AFM tips in the range of sub-10nm tip radius. To this end the authors developed a test vehicle enabling the profile extraction of features calibrated with TEM, thus providing a sound method for the tip-radius calibration. The work can be considered of interest for the community, however major English revisions are required according to this referee in multiple part of the text, especially in the discussion and results. In particular, is also not adviced to refer to the structure as a tip characterizer, maybe the terms "calibration test structure" or grating can be used.
Author Response
Reply to reviewer 4
Dear Reviewers.
Thank you very much for reviewing our manuscript and providing guidance during your busy schedule, which will be of great importance to our future experimental research. We hope that the revised article will meet your requirements and that our independent opinion will be accepted by the reviewers.
The specific responses you are as follows.
Reply to reviewer 4:
Comment 1: The introduction doesn’t provide sufficient background and include all relevant references. It must be improved.
Reply:
Dear Reviewers, thank you very much for your questions on the background section of the introduction and on the references, and here is our response.
Firstly, we have added AFM-IR in the introduction of the article, which is a more enhanced description of the background needs of the article; secondly, we have added new literature in the reference section, and all of them are closely related to AFM-IR. We hope our revision can meet your requirements.
Comment 2: Not all results are clearly presented. It must be improved.
Reply:
Dear reviewers, other reviewers have also made comments and we have revised all the reviewers' comments combined. We attach here the revised version of the manuscript, which we believe is more clearly articulated.
Comment 3: The work of Wu et al. reports on a dedicated structure for the calibration of AFM tips in the range of sub-10nm tip radius. To this end the authors developed a test vehicle enabling the profile extraction of features calibrated with TEM, thus providing a sound method for the tip-radius calibration. The work can be considered of interest for the community, however major English revisions are required according to this referee in multiple part of the text, especially in the discussion and results. In particular, is also not advised to refer to the structure as a tip characterizer, maybe the terms "calibration test structure" or grating can be used.
Reply:
We thank you for the issues raised on the writing of the article, and we have revised and optimized the whole text by checking spelling and grammar issues, taking into account the comments of various reviewers. Second, for the naming of the probe calibrator, we surveyed several papers (Orji, N.G., et al, Ultramicroscopy 2016, 162, 25-34.) ( Villarrubia, J.S., et al, Journal of Vacuum Science & Technology B 1996, 14, 1518-1521.)( Wang, C.M., et al, Journal of Nanoscience and Nanotechnology 2009, 9, 803-808.)( Kwak, G.Y., et al, Applied Surface Science 2021, 565.)( Wang, C. M., et al, Journal of Chinese Electronic Microscopy Society 2007, 26, 576-581.),The related studies all indicate that tip characterizer is a more professional and standard term, but we also think that the reviewer's statement is The reviewer's statement is acceptable.

Round 2
Reviewer 1 Report
It is OK.
Reviewer 3 Report
I do not have more comments
Reviewer 4 Report
The proposed edits have improved the quality of the overall paper now considerable for publication.